# Effects of a Ceiling Fan Ventilation System and THI on Young Limousin Bulls’ Social Behaviour

**DOI:** 10.3390/ani12101259

**Published:** 2022-05-13

**Authors:** Silvia Parrini, Francesco Sirtori, Maria Chiara Fabbri, Aldo Dal Prà, Alessandro Crovetti, Riccardo Bozzi

**Affiliations:** 1Dipartimento di Scienze e Tecnologie Agrarie Alimentari Ambientali e Forestali, Università degli Studi di Firenze, Piazzale delle Cascine 18, 50144 Florence, Italy; silvia.parrini@unifi.it (S.P.); francesco.sirtori@unifi.it (F.S.); alessandro.crovetti@unifi.it (A.C.); riccardo.bozzi@unifi.it (R.B.); 2Institute of BioEconomy-National Research Council (IBE-CNR), Via Giovanni Caproni 8, 50145 Florence, Italy; aldo.dalpra@ibe.cnr.it; 3Centro Ricerche Produzioni Animali—CRPA S.p.A, Viale Timavo 43/2, 42121 Reggio Emilia, Italy

**Keywords:** ceiling fan, Limousin beef cattle, heat stress, social behaviour

## Abstract

**Simple Summary:**

Natural and anthropogenic factors have accelerated climate change. The frequency and intensity of summer heat waves with hot temperature and high humidity are increasing in temperate regions due to global warming. Heat might be very stressful for beef cattle reared in intensive indoor systems, impairing their health and welfare, and practical solutions should be developed and tested for its mitigation. The ceiling fans used in this study are able to slightly modify Limousin beef cattle behaviour, suggesting potential positive effects on animal welfare when ventilation systems are used during the whole heat stress period. Indeed, the number of hours with high temperature humidity index values could determine cumulated stress mainly modifying the animals’ behaviour.

**Abstract:**

The study investigated the relationship between the temperature humidity index (THI) and the behaviour of 24 young fattening Limousin bulls reared in two farms in Tuscany, Italy. In each farm, six animals were undergone to ceiling fans (switched on at THI values up to 72), and six animals represented the control group. The trial lasted three days for two consecutive weeks in August 2020. Behavioural observations were conducted using scan sampling technique and eating, ruminating, drinking, resting and other social activities were registered every 5 min, from 9.30 am to 4.00 pm. Two different microclimatic conditions were evaluated to assess the effect of the ventilation system: normal (THI < 78) and alert (THI ≥ 78) conditions. Results showed that the ventilation system had significant effects increasing inactivity and lying down compared to control groups and decreasing eating and drinking activities. THI alert condition caused a significant decrease in eating and an increase in lying down behaviours. Ventilation system did not influence the animals’ cleanliness. The ceiling fans’ efficiency in changing the behaviour of young fattening bulls was demonstrated but further studies are needed to assess the ventilation system effects, especially during longer heat stress periods.

## 1. Introduction

Natural and anthropogenic factors have accelerated climate change, with global surface temperatures predicted to increase by 1.1–6.4 °C by 2100 [1]. Due to global warming in recent decades, summers in temperate regions have been characterised by adverse weather events in which heat waves alternate with cold spells that negatively affect young bulls’ welfare, productivity and survival [2,3,4,5]. An alarm for beef cattle farmers in Central and Southern Europe could come from the seasonal dynamics of the temperature humidity index (THI) that showed an increasing trend of 0.73 units in summer values [6]. Thermal environments, in terms of air temperature (Ta), relative humidity (RH) and THI, were originally developed to predict heat stress in cattle [7].

THI is widely used to assess thermal stress, the most commonly used formulas generally derive from indices based on the sensation of human thermal comfort [8]. The THI values are in a range between 0 and 100 and for beef cattle they have traditionally been classified as normal (THI < 74), alert (74 ≤ THI < 78) or danger (78 ≤ THI < 84) categories [9]. In normal conditions of intensive farming, with coverage from solar radiation, THI allows a reasonable approximation of the environmental conditions. However, ventilation phenomena could be a contributing factor on heat stress conditions. The young bulls reared in intensive fattening farming, as is usually the case in Continental Europe, are subjected to a heat stress experience when exposed to high THI values due to the inability to maintain homeothermy by effectively dissipating heat [10]. Establishing the real state of welfare of animals is complex, because it depends on aspects such as health, environment and their ability to express natural behaviour [8].

The effects of heat stress on young bulls are negatively correlated with growth performance, dry matter intake [11], animal welfare [12], physiological parameters and plasma thyroid hormone concentration [13]. The inability to find a strategy to defend against hot day impacts causes discomfort, resulting in a negative mental state impacting on biological functioning and subsequent productivity [14]. In this context, the ability to express natural behaviour specifically relative to feeding and dynamic activity, as the time spent eating or lying down, was often studied as the animals’ response to environmental factors [15]. Available studies on the effect of thermal stress and the related mitigation strategies have mainly focused on dairy cows [16,17,18] where the effects on production are verified in a short time [17]. 

Some studies reported the influence and the possible mitigation strategies to manage heat stress on beef cattle including the Limousin breed [5,13]. Strategies to face heat stress include changing the feeding regimen [15], providing additional drinking water points [19], using cooling systems, such as ventilation [5,20] and water sprinkling or misting [21,22] and in outdoor breeding, tree shading is evaluated [23]. The use of water, although useful [24], if not well calibrated, has repeatedly shown negative aspects due to the greater slipperiness of some types of flooring with consequent risk of injury for the young bulls [21]. In addition, greater humidity and the consequent decrease in animal cleanliness, lead to higher costs for more frequent bedding renewal [24]. Moreover, the effectiveness of these evaporative cooling systems decreases when RH increases [25]. As described above, there are multiple management strategies to mitigate heat stress on young bulls reared indoors, but forced ventilation systems without misting or sprinkling represent one of the most used [5]. The availability of studies concerning the relationship between microclimatic changes in the housing conditions of young bulls with their social behaviour and other important aspects for the beef supply chain is limited; among these aspects, the impacts of cleaning during a feeding period were not carefully evaluated in the published literature as other environmental mitigation strategies [26]

The aim of the present study was to assess the effects of ceiling fan ventilation on the behaviour (feeding, dynamic and social activity) and cleanliness of young Limousin bulls kept indoors on deep litter during the finishing period, comparing two THI levels (normal and alert).

## 2. Materials and Methods

### 2.1. Animals, Housing and Management

The study was carried out on two beef cattle farms located in a hilly area of Mugello, Dicomano, Tuscany (Italy). Farm 1 was located at 43°89′45.4′’ N 11°50′63.1″ E; altitude 189 m a.s.l., while farm 2 was at 43°89′06.8″ N 11°50′68.4″ E; altitude 194 m a.s.l.

The study was conducted during the indoor feeding period in August 2020 for three consecutive days, for two weeks in each farm considered. 

In each farm, the herd was composed of 12 young Limousin fattening bulls housed in two pens of six animals:One pen was provided with a ceiling fan ventilation system (V);One pen without ceiling fans as the control treatment (C).

Young bulls were about 15 months old in farm 1 and 15.5 months old in farm 2. Experimental pens were approximately 28 m^2^ where straw for the deep litter was added twice a week in both farms. The pens did not provide restraint systems for the animals, they were equipped with 4 automatic water filling troughs. The feeding was administered along a feeding corridor placed immediately outside the pen where the animals had full access by protruding their heads outside the fence, as is normal practice in similar structures. The animals had no possibility of movement outside the pens. The lighting was provided by neon lamps to support the natural light provided by openings.

Litter samples from each pen were collected each experimental day and the moisture content was measured according to the Association of Official Agricultural Chemists (AOAC) (2016).

Ceiling fans (Big Fan 3m-5p, Arienti & C. SRL, Pieve Fissiraga, LO, Italy) were placed above the V pens and were switched on one month before the first experimental day in order to allow animal adaptation to the operational system. In each farm, pens were arranged in order to prevent air movement in the control pens.

Each ceiling fan was composed of a central impeller (0.25 m height) and five aluminium blades 1.35 m long that created an aerodynamic system 3 m in diameter. The equipment was positioned at a height of approximately 3.5 m from the floor. Each unit ventilated a surface of 8 m in diameter and moved an increasing air volume from 12723 m^3^/h to 37407 m^3^/h under the fan. Power consumption reported on the technical sheet was from 0.12 to 0.24 kW/h. A multifunction electronic control unit managed an automatic switching system thanks to the recording of temperature and humidity via environmental sensors. The system was placed inside the barn and the ceiling fans were checked to see if they worked based on the THI values. The ceiling fans were activated by the system when THI reached a value of 72. Fan speed was measured using a portable thermo anemometer (MS6252A digital anemometer—Proster, Hong Kong) at 2.00 pm during all days of the trial in both V pens of farms 1 and 2.

Furthermore, the indoor micro-environmental condition (temperature and humidity) was measured every 5 min for the entire August month period using four portable hygro-thermometers located at 1 m above the animals. THI was computed using the following formula [27]:(1)THI=1.8×Temperature−(1−(Humidity/100))×(Temperature−14.3)+32

An hourly average THI has been calculated: THI < 78 was considered normal (N) while a THI ≥ 78 was classified as alert (A). 

### 2.2. Diets, Feed Samples and Analyses

Farmers were interviewed about management practices regarding several aspects of the diets intended for the animal herd and feed rations (ingredients, origin, mineralvitamin integration). Young bulls were fed ad libitum by the same total mixed ration (TMR) distributed at 9.00 h AM. The main components of the TMR of both farms were sorghum silage, Italian ryegrass, maize and barley meal, concentrates and soy meal. TMR samples were collected as double sampling weekly in each farm throughout the experimental period for chemical analysis. Samples was oven-dried at 65 °C until constant weight ground to 2 mm and the following parameters were recorded: dry matter, ash, crude protein, neutral detergent fibre with amylase and sodium sulphite method [27] (aNDFom, % of DM), acid detergent fibre (ADF, % of DM), acid detergent lignin (ADL, % of DM), in vitro aNDFom digestibility via 24 h in vitro fermentation (dNDF24 h, % of aNDFom) and undigested NDF after 240 h (uNDF240 h, % of aNDFom [28], starch (% of DM), sugar (% of DM) and net energy for lactation (NEL, kcal kg DM−1) using the instrument Foss NIR-System 5000 monochromator (NIR-System, Silver Spring, MD, USA) according to [29]. Specific composition of the diets (expressed as percentage on DM) as well as chemical composition have been reported in Table 1.

### 2.3. Animals’ Behaviour, Health and Status Cleanliness

Young bull behaviour, health status and cleanliness were evaluated for three consecutive days for two weeks in August. 

Behaviour observation of the animals was carried out by trained observers. A procedure of familiarisation was applied with animals before the trial days in order to avoid observers interfering with the spontaneous animals’ activities. Animals were considered to become familiar with observers when these latter could remain closer to them for about 5 min without affecting or modifying their activity. 

In situ direct observations were conducted using the scan sampling technique [8,31] recording the number of visible young bulls engaged in feeding, dynamic and social activities at predetermined time intervals of 5 min. 

Observations were carried out during daylight hours starting at about 9.30 am and finishing at 4.00 pm. 

Animal activities were grouped into eating, ruminating, drinking, inactivity, lying down, moving, grooming, rubbing and other activities (slipping, exploring, mount and fight). Data were then expressed in minutes within one hour, assuming that each behaviour persisted for the entire 5 min interval. 

Young bulls’ health status was evaluated recording possible disease, medical treatments, lameness and pathogen infections. Animals have been monitored daily and any abovementioned events were showed.

Each young bull was scored for cleanliness after visual inspection by the observers. 

An individual body cleanliness evaluation was carried out analysing four anatomical parts of the body: head, hind limb, tail and ventral/side part adapting the procedure proposed by [2,24]. Cleanliness was scored every day at the same hour (02:00 pm), according to the following levels: 1 = clean with a few small dirt spots at most; 2 = moderately dirty; 3 = mostly covered in dirt. 

### 2.4. Statistical Analysis

Records for behavioural normally distributed data (ruminating, inactivity and lying down) were analysed under the following linear model, using the R function *lm* of *stats* R package [32]:(2)$$Y_{\mathrm{ijkt}}=µ+S_{i}+F_{j}+{\mathrm{THI}}_{k}+\left(S_{i}\times F_{j}\right)+(S_{i}{*\mathrm{THI}}_{k})\times e_{\mathrm{ijkt}}$$
where Y is the observation expressed as minutes/hour for ruminating, inactivity and lying down; µ = the overall mean; S = fixed effect of the ith ventilation system (2 levels: ceiling fans and without ceiling fans); F = fixed effect of the jth farm (2 levels); THI = fixed effect of the kth THI class (2 levels: Alert-THI ≥ 78 and Normal-THI < 78); and e = random residual.

Eating, drinking and grooming events not being normally distributed (W < 90) were normalised using the logarithmic transformation, i.e., ln(Y + 1), and then analysed with the previous described model. 

Rubbing, moving and other activities exhibited a non-parametric distribution and then evaluated with three different one-way non-parametric tests using Kruskal–Wallis test (Proc NPAR1WAY of SAS 9.3; SAS Institute Inc.) [33].

Young bulls’ cleanliness data were analysed under the following linear model, using the R function *lm* of *stats* R package:(3)$$Y_{\mathrm{ijlpt}}=µ+S_{i}+F_{j}+Z_{l}+O_{p}+e_{\mathrm{ijlpt}}$$
where Y is the cleanliness score; µ = the overall mean; S = fixed effect of the ith ventilation system (2 levels: ceiling fans and without ceiling fans); F = fixed effect of the jth farms (2 levels); Z = fixed effect of the lth body zone (4 levels: head, hind limb, tail and ventral/side); O = fixed effect of the pth observer (2 levels); and e = random residual. The least square means were calculated for the aforementioned models with the R *lsmeans* package [34].

## 3. Results

Environmental outdoor conditions of the month of August are reported in Figure 1 as minimum and maximum temperatures as well as average THI values. The trial periods are highlighted in dotted green lines. The most important peak of average THI was recorded in the first days of August. During the trial days, the maximum temperature was always above 30 °C. The highest average THI value was recorded on the first day of observation of the animals. The minimum temperature values referred to the night period when young bulls were not observed, values were always above 20 °C with the exception of the 29 August of the trial days. 

Data of THI within farm and between the ventilation systems were reported as boxplots in Figure 2 for the two different trial periods. 

THI values trends were similar in the two farms: the second week of observations (26–28 August) was surely characterised by more constant and stable THI values than the first one, where average THI highly varies between the different days. 

In both farms, the THI distributions showed the highest values on August 21st, above 80 in farm 1 and close to 80 in farm 2. 

The effect of the ventilation system treatment, THI condition, farm and their interaction on young bulls’ behaviours is reported in Table 2.

Ventilation system affected the feeding and dynamic behaviours of the young bulls. In the control pens, animals spent more time eating (10.70 min/h) than in the ventilated pens (7.46 min/h), and more was time spent drinking by the C group compared to the V group (2.77 vs. 2.03 min/h). No differences were reported for the rumination time ranging between 12.90 to 13.90 min/h. Temporal observations of inactivity were significantly lower in the C group (9.58 min/h) compared to the V group (12.87 min/h). In particular, it seemed that the young bulls under the ventilation system spent more time lying down than the control group. Grooming on oneself or with other young bulls was not affected by the ventilation system. Moving activity did not show differences as reported in Table 2. Contrary, rubbing and other activities mainly linked to traditional social behaviours such as exploring, mounting and fighting were higher for young bulls of the V pens than C pens.

Relative to the THI condition, young bulls subjected to alert THI values spent less time eating (7.46 min/h) than animals advantaged by normal THI conditions (10.70 min/h), while drinking as well as rumination were not significantly different under the two THI conditions. Inactivity behaviour did not differ between the different conditions, while young bulls spent more time lying down during alert than in normal THI conditions (18.10 vs. 13.40 min/h). Furthermore, during alert, THI young bulls spent equal time grooming compared with normal THI values level (2.62 vs. 2.90). Non-parametric behaviours (moving, rubbing and other activities-Table 2) did not show differences based on THI condition. 

Results also revealed that farms had an effect on eating, inactivity, grooming behaviour and rubbing, even if differences between farms were not the object of this study.

The interaction between the ventilation system and the farm condition was significant for inactivity and lying down. In farm 1, C bulls spent less time inactive with respect to the bulls of V pen (6.3 vs. 12.7 min/h) while no differences were found between C and V in farm 2 (12.9 and 13.1 min/h, respectively). Regarding the lying down position in farm 1, animals of the C group spent more time lying than V (16.7 vs. 12.8 min/h), while the opposite occurred in farm 2 (14.8 vs. 18.8, respectively).

Regarding non-parametric data (Table 2), interactions were significant for almost all of the behaviours.

The effect of the ventilation system treatment, farm and anatomical zone on the cleanliness score was reported in Table 3. Similar values were shown for groups C and V, with values between the first and the second level of cleanliness indicating that young bulls were clean or had small dirt spots to moderate dirt cover. Nevertheless, slightly higher values were reported for farm 1 than farm 2, indicating a lower cleanliness level.

An individual body cleanliness evaluation did not show differences among anatomical zones considered with a score ranging between 1.6 and 1.7 which means a high–moderate cleanliness level. 

Finally, the health status of the animals was always excellent and no particular events of nasal discharge, conjunctivitis or lameness were recorded.

## 4. Discussion

This research was carried out in two different farms of the same geographical area and altitude level, rearing the Limousin breed. The Limousin breed has been exported worldwide for production purposes in pure or cross systems [35]. Features such as rugged profile, high casing growth, muscle performance, good power conversion rate and extreme ease of childbirth make Limousin the main French beef breed. This breed is becoming increasingly important day by day in terms of animals. In Italy, enrolled animals in the Limousin Herd Book total 322,321 individuals [36]. Limousin young bulls in the last phase of finishing were considered in two comparable rearing systems housed indoors protected from direct solar radiation and provided with similar TMR and management. The study was carried out in August, where in recent years more difficult conditions were shown due to the lack of ventilation in summer and highly humid environmental conditions. Indeed, over the last decade, climatic changes have also brought the problems of thermal stress in Central Italy. In fact, the area involved in the study is characterized by a hilly environment inserted in the mountain context of the Italian Apennines. In recent years, it was highlighted that the summer months (from June to August) were characterized by an increase in heat waves and high THI values [37].

In farm 2, ventilation system seemed to be more efficient than farm 1 considering that the median of the THI values was lower for the total time of the trial period compared to the control, excluding day 3 of the first week, when there were the highest THI conditions (Figure 2). However, ventilation systems started at the same THI (THI = 72).

Critical THI thresholds for young bulls have been identified by several authors [3,6], despite the fact that the THI index may not be the best indicator because it does not consider the accumulated heat load [38]. 

The influence of the ventilation system on eating has also been reported by [39,40], which argued that animals in conditions of higher heat stress tended to compensate by eating smaller meals more frequently during the day, in order to maintain the daily feed intake. Relative to the ventilation system, it must be considered that TMR was administered before the starting of the observation and the young bulls of the V pens seemed hungrier, probably because they were less affected by the effect of accumulated stress. Curtis et al. [41] reported no significant correlation between DMI and the daily environmental conditions but reported a 3 to 5 d of delay in DMI response. Cattle may accumulate heat during the day (body temperature rises) and dissipate the heat at night. If there is insufficient night cooling, cattle may enter the following day with an ‘accumulated’ heat load [38]. An ad libitum distribution of TMR could be a strategy to mitigate heat stress as reported by some authors [42]. Even if the feed intake is not evaluated in our study, some authors suggested the strategies of the TMR distribution during the cooler parts of the day in particular during the late afternoon, 2 to 4 h after peak temperature [9,43], in order to alleviate some of the heat load from the feed. 

Animals of the V pens spent less time drinking probably because they had less of a need to reduce their core body temperature, and the longer time spent drinking by young bulls of the control group could represent attempts to balance heat loads already reported for ruminants reared in grazing systems [44]. The higher number of events in the drinking zone probably coincided with a greater water consumption as reported by Magrin et al. [5], who described higher water ingestion (l/d) in the control thesis (no ceiling fans), implemented by animals to promote animal heat dissipation.

Resting behaviour including inactivity and lying down had a different pattern trend based on the farm. The expected results reflected the pattern of farm 1 where the C group spent less time inactive and lying down compared to V probably because standing helps increase heat loss [45]. On the contrary, in farm 2, ventilated young bulls spent more time lying down than C, according to other studies [17,46] which reported more lying down time in a no stress condition than under heat stress conditions in grazing animals [46]. 

The total time spent carrying out other activities and rubbing was very short or absent during the days in both groups (V and C). However, young bulls of the V group dedicated more time to social or specific activity (rubbing, exploring, mount), that is typical of cattle ethogram behaviour and could indicate a higher welfare condition. Nevertheless, these activities only provide information about social dynamics in particular environments and were not included in the majority of studies due to their lower frequency [17]. 

During THI alert condition, animals spent less time eating and exhibited no feed selection behaviour. According to our study, Brown-Brandl et al. [9] reported that above a threshold of THI that the authors identified as the “emergency category” (higher than our alert THI), the total duration spent eating decreases as well as daily feed intake, number of meals and meal size. On the contrary, below the heat stress limit, as reported in the comparison between ventilation treatments, animals seemed to increase the number of meals and the meal size to compensate for the DMI intake. Numerous research studies suggested the decrease in DMI intake during heat stress a as factor in maintaining core body temperature [40] which could be included in the mechanism of acclimation and adaptation to hot environments, although many of the changes in metabolic pathways are not yet understood [46,47]. Even if the total time spent eating could be linked to a daily specific period or could be affected by accumulated stress, a decreased interest in feed includes a voluntary reduction in DMI that is never favourable both to animal performance and wellbeing [40].

The absence of difference in rumination time or specifically chewing activity suggested that young bulls did not select specific dimensional fraction of TMR or reduced forage/concentrate ratio depending on the tested theses. The same time engaged by young bulls in rumination during different THI conditions could also be associated with the fact that the activity of rumination is mainly performed after sunset and during the night, as previously reported in young Limousin bulls reared in an intensive system [38]. Nevertheless, other authors reported that in an attempt to reduce metabolic heat and safeguard their homeothermy [4], animals spent less time ruminating in THI alert condition [5]. 

THI alert conditions seem to not affect the time spent drinking according to Magrin et al. [5] which reported that the number of visits to the drinkers was not affected by any of the effects considered (THI condition and ventilation system) even if an higher water intake was recorded during alert THI periods. Usually heat stress is associated with an increased water consumption (data not observed in our study), but the time spent drinking could also be linked to a cooling sensation in staying near the water. Furthermore, higher water consumption has been more frequently reported in the long period of heat stress in the summer season compared to winter [48]. However, numerous factors could affect the absence of difference in this behaviour between environmental conditions or physiological functions including: regulation of core body temperature, growth and development, composition of TMR, digestion and metabolism [40,48].

Despite the lack of differences in observations of inactivity behaviour, the increased time spent lying down during the THI alert period is in agreement with Sullivan et al. [49] but in opposition to the theory of Ansell [45], who found that standing time usually increased in heat-stressed cattle. 

Regarding cleanliness, it seemed that young bulls achieved slightly better load in farm 2 compared to farm 1; nevertheless, multiple factors linked to microclimate, environmental and structural conditions can affect these different levels of cleanliness. This slight difference seemed to be linked to THI indoor trend during the trial days when a more efficient functioning of the ceiling fan was reported in the environmental situations of farm 1. Independent from the ventilation system or body zone, the young bulls in our trials achieved medium–high cleanliness levels that are favourable for integument health, hoof health [50], overall welfare status [15] and mandatory at slaughterhouses. 

Finally, numerous factors affected the wellbeing and the response of animals to heat stress conditions as well as to the effectiveness of mitigation strategies such as ventilation systems. Cattle may accumulate heat during the day and have insufficient night cooling when the ventilation system is not “on” and have specific behavioural responses as a result of cumulative heat load [38]. Furthermore, as suggested by Magrin [5], the benefits observed by the ceiling fans in young fattening bulls might be more evident under severe heat stress conditions. Nevertheless, unlike small ruminants, where synergies between various stressors have been detected, the impact of multiple stressors linked to heat stress has not been sufficiently investigated in cattle [40] as well as the effect of cumulative heat stressors [38].

## 5. Conclusions

The results obtained in the study showed slight differences between the experimental factors taken into consideration. The spatial location of the farms probably greatly influenced the result as it is not characterized by extreme phenomena (long consecutive period of heat stress) in which the mitigation systems could show greater success as reported by various bibliographic works cited. Ceiling fans used in this study are able to slightly modify Limousin beef cattle behaviour, suggesting potential positive effects on animal welfare, probably more effective if a ventilation system is used during the entire period of thermal stress and in more extreme stressful situations. The ventilation system did not affect the cleanliness of animals; however, further studies appear necessary to evaluate the effects of the ventilation system, particularly over longer periods of heat stress when cumulative heat stress and multiple stressors could be the key factor.

## Figures and Tables

**Figure 1 animals-12-01259-f001:**
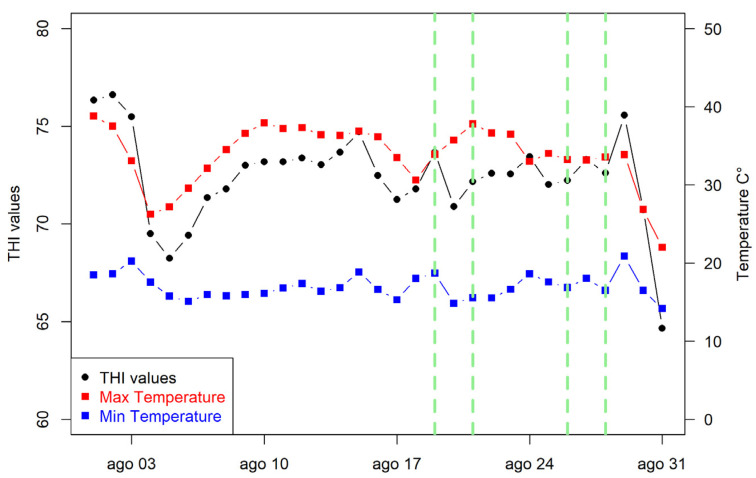
Outdoor minimum, maximum temperatures and temperature humidity index (THI) values recorded throughout August 2020. The study periods are within the green dotted lines.

**Figure 2 animals-12-01259-f002:**
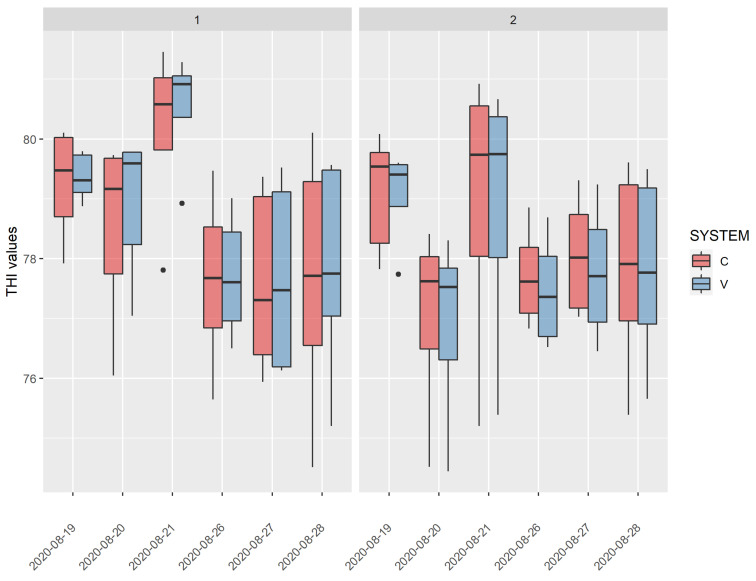
THI index distribution during the study period within the farms (1 and 2) and treatment (C and V).

**Table 1 animals-12-01259-t001:** Components of the diet and chemical composition of TMRs.

Item	Farm 1	Farm 2
TMR component		
Sorghum silage, % of DM	1.8	3.2
Alfalfa, % of DM	2.0	1.5
Italian ryegrass, % of DM	2.0	1.0
Maize and barley meal, % of DM	1.8	4.5
Concentrates, % of DM	2.5	0.8
Soy meal, % of DM	0	0.5
Chemical composition		
DM, %	54.8	56.4
Ash, % of DM	8.8	9.3
CP, % of DM	11.7	11.6
aNDFom, ^1^ % of DM	45.4	29.2
ADF, % of DM	30.9	17.9
ADL, % of DM	5.2	2.7
dNDF24 h, ^2^ % of aNDFom	40.0	35.6
uNDF240 h, ^3^ % of aNDFom	17.0	15.5
Starch, % DM	12.9	32.3
Sugar, % DM	4.8	2.9
Fat, % DM	2.1	2.5
(NEL) ^4^, kgcal/kg DM	1.356	1.496

^1^ aNDFom = amylase and sodium sulphite-treated NDF with ash correction; ^2^ dNDF24 h = In vitro aNDFom digestibility via 24 h in vitro fermentation; ^3^ uNDF240 h = undigested NDF estimated via 240 h in vitro fermentation; ^4^ Net energy of lactation, according to NRC [30].

**Table 2 animals-12-01259-t002:** Effect of ventilation system treatment, temperature humidity index (THI) condition, farm and their interaction on young bulls’ behaviour expressed as minutes/hour.

Behaviour (min/h)	Ventilation System			THI			Farm *p*-Value	Ventilation System × Farm *p*-Value	Ventilation System × THI *p*-Value
	C	V	*p*-Value	A	N	*p*-Value			
Eating ^1^	10.70	7.46	0.023	7.46	10.70	0.027	0.014	ns	ns
Ruminating	13.90	12.90	ns	14.80	12.01	ns	ns	ns	ns
Drinking ^1^	2.77	2.03	0.002	2.34	2.42	ns	ns	ns	ns
Inactivity	9.58	12.87	0.001	11.00	11.40	ns	<0.001	0.010	ns
Lying down	15.70	15.80	0.041	18.10	13.40	0.010	ns	0.029	ns
Grooming ^1^	2.86	2.63	ns	2.62	2.90	ns	0.005	ns	ns
Moving ^2^	1.08	1.93	Ns	1.35	1.70	ns	ns	<0.001	<0.001
Rubbing ^2^	0.77	1.15	0.041	1.06	0.81	ns	<0.001	0.01	ns
Other ^2^ activities	0.29	1.24	<0.001	0.60	0.98	ns	ns	<0.001	<0.001

V = group with ceiling fan ventilation system; C = control group without ceiling fan ventilation system; A = alert THI condition > 78; N = normal THI condition < 78; ns (no significant) = *p*-value > 0.05. ^1^ Data processed after logarithmic transformation using the formula ln(Y + 1) and presented as a back-transformed average. ^2^ Non-parametric data evaluated with three different one-way non-parametric tests using Kruskal–Wallis test.

**Table 3 animals-12-01259-t003:** Effect of ventilation system treatment, farm and body zone on cleanliness. Values ranged from 1 to 3 (1 = clean with a few small dirt spots at most; 2 = moderately dirty; 3 = mostly covered in dirt).

Effect	Cleanliness		
			*p*-Value
Ventilation system	C	1.66	ns
	V	1.59	
Farm	1	1.71	0.02
	2	1.54	
Zone	Head	1.60	ns
	Hind limb	1.73	
	Tail	1.60	
	Ventral and side part	1.60	

V = group with ceiling fan ventilation system; C = control group without ceiling fan ventilation system; ns (no significant) = *p*-value > 0.05.

## Data Availability

Data are available on request by the corresponding author.

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
