# Peer review of "Effects of a Ceiling Fan Ventilation System and THI on Young Limousin Bulls’ Social Behaviour"

_animals, 2022, doi:10.3390/ani12101259_

Round 1

Reviewer 1 Report

The article is interesting from a clinical point of view, but it has several issues to solve before publication.

The major weakness is the study population: reaching any conclusion with 6 bull per pen (2 farms) is quite complicated. Moreover, the variable ‘farm’ can disturb the results.

The introduction must be rewritten. Suddenly, in the objectives, behaviors and cleanliness appear. No references of the effects of heat stress on these parameters are explained.

Another revision to be done:

Please, eliminate figure 1

Authors write very short paragraphs, especially in the material and methods and discussion sections. Please, take into account that a paragraph is ‘a piece of writing, usually dealing with a single theme’.

I don’t think that figure 2 and 3 are needed. Maybe with only one we can manage.

Please modify conclusions. A conclusion is something irrefutable

Author Response

Reviewer 1

Comments and Suggestions for Authors

The article is interesting from a clinical point of view, but it has several issues to solve before publication.

The major weakness is the study population: reaching any conclusion with 6 bull per pen (2 farms) is quite complicated. Moreover, the variable ‘farm’ can disturb the results.

Farm effect must be considered as fixed effect to correct the model.

The introduction must be rewritten. Suddenly, in the objectives, behaviors and cleanliness appear. No references of the effects of heat stress on these parameters are explained.

Authors thank for the suggestion; the introduction has been expanded with reference related to behaviour and cleanliness status. These aspects are however little investigated.

Another revision to be done:

Please, eliminate figure 1

Figure 1 has been deleted.

Authors write very short paragraphs, especially in the material and methods and discussion sections. Please, take into account that a paragraph is ‘a piece of writing, usually dealing with a single theme’.

Authors think that short paragraph are generally more comprehensible, however some changes have been done.

I don’t think that figure 2 and 3 are needed. Maybe with only one we can manage.

In the new version figure 2 is become figure 1 and figure 3 is figure 2. Authors think that these figures must be separated because figure 1 represents outdoor THI index and temperature trend, while Figure 2 is  essential to highlighting the effect of ceiling fans on THI index within farm.

Please modify conclusions. A conclusion is something irrefutable

Conclusions have been rewritten.

Reviewer 2 Report

The manuscript animals-1691921 falls within the scope of the journal. Although the study is interesting still lots of clarifications are needed before the paper is considered for publication. The authors must address the following points

  • The technical program lacks clarity. Authors have to justify the question of why only three consecutive days were chosen for the study trial and why not the entire 14 days as mentioned in the study.
  • Authors are requested to properly mention the difference between both the groups in the study.
  • Describe the methodologies of recording all the variables in the study
  • Authors are suggested to provide clear photographs of both pens provided with a ceiling fan ventilation system (V) and without a ceiling fan as a control treatment (C).
  • Authors are suggested that all the abbreviations should be expanded both in abstract and in-text and thereafter it should be consistently abbreviated.
  • It is suggested to merge Table 2 and Table 3 and leave a footnote below the table if in case needed.
  • It is suggested for the authors describe the breed involved in the study and its economic significance to the local farmers of its origin.

Author Response

Reviewer 2

Comments and Suggestions for Authors

The manuscript animals-1691921 falls within the scope of the journal. Although the study is interesting still lots of clarifications are needed before the paper is considered for publication. The authors must address the following points

  • The technical program lacks clarity. Authors have to justify the question of why only three consecutive days were chosen for the study trial and why not the entire 14 days as mentioned in the study.

The authors report in the manuscript that the study of behaviours is done for 3 consecutive days replicated in two consecutive weeks and not for 14 days. The experimental design involved the behavioural analysis for 3 consecutive days repeated in 2 replicates. The choice to carry out the replicas in consecutive weeks was to remain within a similar period for both weeks from the climatic point of view, having the possibility of recording a greater amount of data. The choice to carry out the study for only three consecutive days was based on similar references, as in Magrin et al. (2016) in which the days were 4 but without replicas.

  • Authors are requested to properly mention the difference between both the groups in the study.

Authors don’t understand what differences reviewer want to highlight

  • Describe the methodologies of recording all the variables in the study

Behaviour observation methodology was carried out according to Martin et al., (2007) and Altmann (1974) as reported in the manuscript. Cleanliness method was improved at L183. Young bulls’ health status meaning has been described (L177).

Martin, P.; Bateson, P.P.G. Measuring Behaviour: An Introductory Guide; 3rd ed.; Cambridge University Press: Cambridge; New York, 2007; ISBN 978-0-521-82868-0

Altmann, J. Observational Study of Behavior: Sampling Methods. Behaviour 1974, 49, 227–266, doi:10.1163/156853974X00534.

  • Authors are suggested to provide clear photographs of both pens provided with a ceiling fan ventilation system (V) and without a ceiling fan as a control treatment (C).

Control and ventilated pens were placed in separate places of the barns to prevent air movement in the control pens, so it’s impossible to take a single picture that can shows the entire barn.

Authors are suggested that all the abbreviations should be expanded both in abstract and in-text and thereafter it should be consistently abbreviated.

Abbreviations have been checked (e.g., L44, L133)

  • It is suggested to merge Table 2 and Table 3 and leave a footnote below the table if in case needed.

Done

  • It is suggested for the authors describe the breed involved in the study and its economic significance to the local farmers of its origin.

Done (L295-300)

Reviewer 3 Report

Regardless of the formulation of the general purpose of the research / study, it would also be worth writing in the article what was the cognitive (scientific) goal of the research, and what utilitarian (useful) goals were formulated by the Authors? In my opinion, the review of the state of knowledge in the Introduction chapter could be summarized by the formulation of the research problem. You can write: "The research problem is ...". The research problem can be associated with the indication of a gap in the current state of knowledge regarding the relationship between microclimatic changes in the housing conditions of beef cattle with their behaviour and other features.

In Figure 1, on the bottom map it would be possible to mark (for example with dots) the exact locations of the farms included in the research.

It would be useful to write what part of the year (how many days) covers the period with extreme microclimatic conditions and extreme THI in which fattening cattle are kept in livestock buildings in the region where the research was carried out.

I think that in the Materials and Methods chapter, it would be worth including photos of fans installed above the cattle pens. Regardless of the description of the technical and functional characteristics of the fans, it would be of great help in identifying the details related to the design of the fans and their operation.

Were the fans equipped with automatic control systems (on and off) if they were switched on with a THI of 72? It would be worth writing something more about such a system (automatic control) in the section describing fans.

I do not understand the information provided in the paragraph, lines 114-116. In this paragraph, the authors wrote that the fan speed was measured in both test areas, ie V and C. Meanwhile, area C, as shown by the information in line 95, was a fanless area (as a control area). Please explain this inaccuracy. Moreover, in the lines (95 and 115) the authors use different words, i.e. "pen" and "box(es)" for the same place (research area), which creates unnecessary confusion. It is worth standardizing the vocabulary (in this case the key vocabulary) so that there is no ambiguity when reading and analyzing the text.

In the description of the research objects, it would be worth providing even more information on the housing system for beef cattle. I guess it was a loose housing system? How was animal feeding organized (feeding alley information)? How did the animals use the water? Were there individual or collective drinkers (drinking bowls) in pens? What was the lighting system in the livestock buildings? Did the animals have access to the outdoor areas of the livestock building? By providing such information, it will be possible to indicate that the basic animal welfare requirements have been met.

Were the animals weighed before and after the experiment? I think that such information would make a valuable contribution to the discussion of the research results.

The equation in line 121 could be labeled, for example, (1) or (eq. 1). In the aforementioned equation, I do not agree with the parentheses used to link the relevant quantities with each other. I figured there are four left and six right parentheses. Hence my doubts about the correctness of the parentheses used.

Question regarding line 119 information: At what height were the hygro-thermometers placed above the animals? In my opinion, this is important information, because young animals (and especially cattle) are very interested in contact with the new equipment infrastructure (in this case, measuring devices), which requires appropriate planning of the location of the meters.

Was it not better to use the option of recording animals with cameras and subsequent analysis of the recorded material instead of observers of beef cattle behaviour? Then there would be no disturbances in the herd of animals resulting from the presence of people in the vicinity of the research (experiment) zone. What is the authors' justification for such an approach to the research, using the direct participation of observers to collect data on the behaviour of animals in pens?

Of course, in Tables 2, 3 and 4 I can guess that the abbreviation "ns" means no significant, but it would be worth writing about it somewhere in the text.

Author Response

Reviewer 3

Comments and Suggestions for Authors

Regardless of the formulation of the general purpose of the research / study, it would also be worth writing in the article what was the cognitive (scientific) goal of the research, and what utilitarian (useful) goals were formulated by the Authors? In my opinion, the review of the state of knowledge in the Introduction chapter could be summarized by the formulation of the research problem. You can write: "The research problem is ...". The research problem can be associated with the indication of a gap in the current state of knowledge regarding the relationship between microclimatic changes in the housing conditions of beef cattle with their behaviour and other features.

A sentence has been added

In Figure 1, on the bottom map it would be possible to mark (for example with dots) the exact locations of the farms included in the research.

Figure 1 has been deleted as requested by another reviewer

It would be useful to write what part of the year (how many days) covers the period with extreme microclimatic conditions and extreme THI in which fattening cattle are kept in livestock buildings in the region where the research was carried out.

Details have been added (L305-308)

I think that in the Materials and Methods chapter, it would be worth including photos of fans installed above the cattle pens. Regardless of the description of the technical and functional characteristics of the fans, it would be of great help in identifying the details related to the design of the fans and their operation.

The authors have added some specifics of how the technology works and reporting what the supplier firm has granted.

Were the fans equipped with automatic control systems (on and off) if they were switched on with a THI of 72? It would be worth writing something more about such a system (automatic control) in the section describing fans.

Sentence added (L120-126)

I do not understand the information provided in the paragraph, lines 114-116. In this paragraph, the authors wrote that the fan speed was measured in both test areas, ie V and C. Meanwhile, area C, as shown by the information in line 95, was a fanless area (as a control area). Please explain this inaccuracy. Moreover, in the lines (95 and 115) the authors use different words, i.e. "pen" and "box(es)" for the same place (research area), which creates unnecessary confusion. It is worth standardizing the vocabulary (in this case the key vocabulary) so that there is no ambiguity when reading and analyzing the text.

The sentence has been checked (L128).

A standardization of vocabulary has been done (L128)

In the description of the research objects, it would be worth providing even more information on the housing system for beef cattle. I guess it was a loose housing system? How was animal feeding organized (feeding alley information)? How did the animals use the water? Were there individual or collective drinkers (drinking bowls) in pens? What was the lighting system in the livestock buildings? Did the animals have access to the outdoor areas of the livestock building? By providing such information, it will be possible to indicate that the basic animal welfare requirements have been met.

Sentence has been modiefied  (L120-124)

Were the animals weighed before and after the experiment? I think that such information would make a valuable contribution to the discussion of the research results.

the animals were not weighed to avoid external influences by the operators and because the aim of the study was not on production fluctuation during THI levels.

The equation in line 121 could be labeled, for example, (1) or (eq. 1). In the aforementioned equation, I do not agree with the parentheses used to link the relevant quantities with each other. I figured there are four left and six right parentheses. Hence my doubts about the correctness of the parentheses used.

The formula has been modified but authors prefer to not include eq.1 etc. because only three equations were used.

Question regarding line 119 information: At what height were the hygro-thermometers placed above the animals? In my opinion, this is important information, because young animals (and especially cattle) are very interested in contact with the new equipment infrastructure (in this case, measuring devices), which requires appropriate planning of the location of the meters.

The sentence has been improved (L132)

Was it not better to use the option of recording animals with cameras and subsequent analysis of the recorded material instead of observers of beef cattle behaviour? Then there would be no disturbances in the herd of animals resulting from the presence of people in the vicinity of the research (experiment) zone. What is the authors' justification for such an approach to the research, using the direct participation of observers to collect data on the behaviour of animals in pens?

The choice to register directly in the company was decided for various aspects: first of all, the impossibility through planning to purchase the video surveillance equipment; second reason, the assessment of the cleanliness and health of the animals would have been difficult through video images.  Finally, the direct presence of the operator allows to better record an action when the animal is being picked up by another, given the possibility of moving around this one.

Of course, in Tables 2, 3 and 4 I can guess that the abbreviation "ns" means no significant, but it would be worth writing about it somewhere in the text.

Done

Round 2

Reviewer 2 Report

The authors addressed the queries satisfactorily. The manuscript would benefit if two separate photographs of control & experimental

Author Response

Authors know that the manuscript would benefit with photographs of control & experimental pens, but authors haven't professional photos which can be published on the journal. In addition,  ceiling fans were hired in this project, so it is impossible now to take new photos.